# Dynamic Hyperinflation While Exercising—A Potential Predictor of Pulmonary Deterioration in Cystic Fibrosis

**DOI:** 10.3390/jcm12185834

**Published:** 2023-09-08

**Authors:** Einat Shmueli, Yulia Gendler, Patrick Stafler, Hagit Levine, Guy Steuer, Ophir Bar-On, Hannah Blau, Dario Prais, Meir Mei-Zahav

**Affiliations:** 1Graub Cystic Fibrosis Center, Pulmonary Institute, Schneider Children’s Medical Center of Israel, 14 Kaplan Street, Petah Tiqwa 49202, Israel; pstafler@hotmail.com (P.S.); hagitlevine@gmail.com (H.L.); kids.lung@gmail.com (G.S.); ophirbo@gmail.com (O.B.-O.); hannahblau@hotmail.com (H.B.); prais@tauex.tau.ac.il (D.P.); meir_zahav@clalit.org.il (M.M.-Z.); 2Faculty of Medicine, Tel Aviv University, Tel Aviv 69978, Israel; 3Department of Nursing, School of Health Sciences, Ariel University, Ariel 40700, Israel; yuliagendler@gmail.com

**Keywords:** cystic fibrosis, cardiopulmonary exercise test, pulmonary function, dynamic hyperinflation

## Abstract

Background: Lung function deterioration in cystic fibrosis (CF) is typically measured by a decline in the forced expiratory volume in one second (FEV_1_%), which is thought to be a late marker of lung disease. Dynamic hyperinflation (DH) is seen in obstructive lung diseases while exercising. Our aim was to assess whether DH could predict pulmonary deterioration in CF; a secondary measure was the peak VO_2_. Methods: A retrospective study was conducted of people with CF who performed cardiopulmonary exercise tests (CPETs) during 2012–2018. The tests were classified as those demonstrating DH non-DH. Demographic, genetic, and clinical data until 12.2022 were extracted from patient charts. Results: A total of 33 patients aged 10–61 years performed 41 valid CPETs with valid DH measurements; sixteen (39%) demonstrated DH. At the time of the CPETs, there was no difference in the FEV_1_% measurements between the DH and non-DH groups (median 83.5% vs. 87.6%, respectively; *p* = 0.174). The FEV_1_% trend over 4 years showed a decline in the DH group compared to the non-DH group (*p* = 0.009). A correlation was found between DH and the lung clearance index (LCI), as well as the FEV_1_% (r = 0.36 and *p* = 0.019 and r = −0.55 and *p* = 0.004, respectively). Intravenous (IV) antibiotic courses during the 4 years after the CPETs were significantly more frequent in the DH group (*p* = 0.046). The peak VO_2_ also correlated with the FEV_1_% and LCI (r = 0.36 and *p* = 0.02 and r = −0.46 and *p* = 0.014, respectively) as well as with the IV antibiotic courses (r = −0.46 and *p* = 0.014). Conclusions: In our cohort, the DH and peak VO_2_ were both associated with lung function deterioration and more frequent pulmonary exacerbations. DH may serve as a marker to predict pulmonary deterioration in people with CF.

## 1. Introduction

Cystic fibrosis (CF) is the most common life-limiting autosomal recessive disease, with lung disease being the most significant cause of morbidity and mortality [1].

The loss of lung function is usually measured by the decline in the forced expiratory volume in one second (FEV_1_%) as measured by spirometry. It is undisputed that FEV_1_% decline is a late marker of lung disease in CF [1,2]. Markers of early lung disease are essential, as they indicate a critical time window for intervention with more intensive therapies and the attenuation of the disease trajectory.

Exercise-related parameters are more sensitive than measurements at rest in assessing respiratory health, and, indeed, exercise tolerance is reduced in people with CF [2,3,4].

Exercise performance and respiratory function while exercising was suggested as a prognostic factor over 30 years ago [5]. While later studies failed to prove the superiority of aerobic capacity over the FEV_1_% [6], a recent review concluded that low levels of peak VO_2_ are associated with an increase of 4.9-fold in the risk of mortality in subjects with CF, and it could be an important variable to follow in addition to the FEV_1_% [7]. Other parameters of exercise tests were not extensively investigated.

Inspiratory capacity (IC) tends to increase in healthy people while exercising, as recruited expiratory muscles ensure increased expiratory flow and decreased end expiratory lung volume. Dynamic hyperinflation (DH) is a phenomenon which appears in obstructive lung diseases such as chronic obstructive pulmonary disease (COPD) and CF while exercising. The collapse of the airways during forced expiration in these obstructive lung diseases contributes to lung hyperinflation [8]. Due to airway obstruction, exhalation may not be complete at the time the next breath is initiated, thereby leading to increasing amounts of trapped air at the end of exhalation and increasing end expiratory lung volume. This process is termed dynamic hyperinflation.

Measuring inspiratory capacity is a surrogate measure of dynamic hyperinflation and increased end expiratory lung volume in obstructive lung disease. Assuming that the total lung capacity remains constant, as exercise progresses and the end expiratory lung volume increases, the inspiratory capacity decreases [8,9].

Dynamic hyperinflation can limit ventilation during exercise. The work of breathing is increased, as the inspiratory muscles are at a mechanical disadvantage due to length–tension effects. At first, increased respiratory effort results in increased tidal volume. However, as exercise proceeds, there is a progressive decrease in the IC without a further increase in the tidal volume. This results in the cardinal symptom of dynamic hyperinflation, which is dyspnea upon exertion.

In healthy individuals, the exercise tidal flow volume loop (extFVL) and the maximal flow volume loop (MFVL) are distinct. However, as end expiratory lung volume increases with dynamic hyperinflation, there is an encroachment of the extFVL on the MFVL.

Previous studies have demonstrated a correlation between DH, reduced lung function, and exercise intolerance in CF, but the relationship between DH and subsequent lung function deterioration has not been examined [10,11].

Cardiopulmonary exercise tests (CPETs) are performed periodically as a routine follow-up at our center by people with CF who have an FEV_1_% greater than the 30% that is predicted. The test is performed while clinically stable.

We hypothesized that DH and other exercise parameters might be sensitive markers for predicting lung disease severity and deterioration in people with CF.

## 2. Methods

### 2.1. Patients

We conducted a retrospective longitudinal study that included children and adults with CF (FEV_1_% > 30% predicted) who were followed at the Graub Cystic Fibrosis Center at Schneider Children’s Medical Center of Israel during 2012–2018.

Patients’ demographic and clinical data until 12.2022 were collected, including age, gender, body mass index (BMI), genotype, pancreatic sufficiency status, the presence of CF-related diabetes and CF-related liver disease, the presence of chronic infection with Pseudomonas aeruginosa, pulmonary function tests, courses of intravenous antibiotics (IV) for pulmonary exacerbations, and CFTR modulator treatment status.

Routine medications used by the patients in our CF center include hypertonic saline and dornase alfa inhalations, as well as inhaled and oral antibiotics, in accordance with bacteria found in sputum cultures. Routine treatment also includes physiotherapy, pancreatic enzyme replacement therapy, and vitamins. As previously mentioned, some patients received CFTR modulator treatment during the study period.

### 2.2. Pulmonary Function Tests

Pulmonary function was measured at the lung function laboratory of the hospital’s pulmonary institute according to the American Thoracic Society/European Respiratory Society standards [12]. Spirometry and plethysmography measurements were performed using a constant-volume body plethysmograph (smart PFT body, Medical Equipment Europe GmbH, Hammelburg, Germany). Nitrogen multiple breath washout was performed with the use of an ultrasonic flow meter (smart PFT nebulizer, Ecomedics, Hammelburg, Germany).

Cardiopulmonary exercise tests (CPETs) are routinely performed periodically at our center by people with CF while they are clinically stable and have a FEV_1_% percent predicted (FEV_1_%pp) to be greater than 30% predicted.

Cardiopulmonary exercise testing was performed on an electronically braked cycle ergometer and a ZAN600 computerized metabolic system (variable impedance pneumotachometer sensor technology, ZAN-Messgerate GmbH, Hammelburg, Germany) using a graded protocol, with work rate increasing progressively at 10–20 W/min, depending on the patient predicted values. The work rate was increased with the goal of participants reaching maximal effort within 10–12 min. Pulmonary gas measurements were recorded during exercise and included oxygen uptake, carbon dioxide production, and minute ventilation using a metabolic gas analyzer (MetaLyzer^®^3B, Cortex Biophysik GmbH, Hammelburg, Germany).

IC was measured by spirometry as patients were instructed to inspire fully after normal expiration and then to expire maximally. The volume measured was defined as IC. In order to practice the maneuver, the patients performed several IC maneuvers before commencing the CPET. In addition, during the CPET, maneuvers were repeated every 2–3 min and only technically acceptable tests at maximal exercise, as reviewed by an experienced technician, were included. The change in IC from rest to peak exercise was calculated (IC exercise − IC rest = ∆IC) and DH was defined as a decrease of ≥5% in IC during maximal exercise [13]. Patients were categorized as those with and without evidence of DH. In the absence of expiratory flow limitation, vital capacity and a lack of inspiratory capacity might be the actual limit for tidal volume expansion. It should therefore be noted that expiratory flow limitation was a major feature in our patient group. For this population, as inspiratory capacity is a surrogate measure for end expiratory lung volume, we chose to use this measure in our study.

The criteria for acceptable CPET results included-a respiratory exchange ratio (RER) of ≥1.05 for adults and ≥1.03 for children [14]; peak heart rate > 100% predicted in adults or ≥195 bpm in children [15]; ventilation at peak exercise exceeding maximum voluntary ventilation (MVV) [16]; and VO_2_ peak ≥ 100% predicted or maximal work rate ≥ 100% predicted [14].

### 2.3. Parameters Evaluated

Patients’ demographic data and clinical data were compared between patients demonstrating DH and those without DH. The number of IV antibiotic courses for each patient in the 4 years after the CPETs were also compared between the two groups. FEV_1_% values were measured in the two and four years after the tests were extracted. Lung clearance index (LCI) measurements of all patients within 4 years of the CPETs were also reviewed. Values of FEV_1_% and LCI were compared between DH and non-DH patients.

The primary outcome measures were the rate of FEV_1_% decline in the DH as compared to the non-DH group, the LCI value measured during the 4 years after the CPETs, and the number of pulmonary exacerbations measured by IV antibiotic courses in the 4 years after the CPET in both groups.

The secondary outcome measures were the correlation between peak VO_2_ and FEV_1_%, as well as the LCI values and the number of IV antibiotic courses in the 4 years after the CPETs.

### 2.4. Statistics

The Shapiro–Wilk test was used for assessing normal distribution (*p* > 0.05 indicated that the sample distribution does not deviate significantly from a normal distribution). Data are presented as means (SDs) or median [min, max] for continuous variables and frequencies (%) for categorical variables. Primary outcomes: The differences in CPET parameters between DH and non-DH groups were analyzed using independent samples *t*-tests or Chi test [2] or Fisher’s exact test for smaller samples. Correlations between ∆IC and LCI/FEV_1_% were analyzed using Pearson test, and linear regression models were fitted. The correlation between peak VO_2_ and number of IV courses were analyzed using Spearman test. Trends in FEV_1_% changes over the study period were analyzed using a univariate general linear model. The analysis was performed using IBM SPSS version 29 (Armonk, NY, USA); statistical significance was set at *p* < 0.05.

## 3. Results

During the study period, 61 CPET studies were conducted, of which 20 were technically unacceptable—either due to submaximal effort during the test or due to malperformance of the DH maneuver, and they were therefore excluded. Valid DH measurements were available for 41 tests, which were performed by 33 patients (46% females). Their median age at the time of performing the test was 24.4 years (range 10–61, Table 1).

### 3.1. Baseline Characteristics of the Study Groups

#### 3.1.1. Clinical Parameters—DH vs. Non-DH

While preforming IC maneuvers in rest and at peak exercise, DH was demonstrated in 16 (39%) tests; the mean (range) ∆IC was −14.7% (−6–(−29%)), whereas 61% (25) of the patients did not demonstrate DH, and their mean (range) ∆IC was +9.1% (−3.5–(+40.7%)).

The demographic parameters were similar in the DH and non-DH groups, apart from gender, wherein there were more females in the DH group (69% vs. 32%, *p* = 0.03) (Table 1).

The clinical characteristics of the DH and non-DH groups were similar at the baseline, including the respiratory status, with a median predicted FEV_1_% of 83.5% vs. 87.6%, respectively (*p* = 0.174).

#### 3.1.2. Cardiopulmonary Measures—DH vs. Non-DH

Comparison of CPET parameters between DH and non-DH groups demonstrated a significantly higher breathing frequency in the DH group (mean (SD), 143.8 (34.6) vs. mean (SD) of 121.4 (30.6); *p* = 0.042). Other parameters, including the breathing reserve, peak VO_2_%, and VO_2_%, that were predicted at the anaerobic threshold were similar between the two groups (Table 2).

### 3.2. Prognostic Values of CPET Parameters

#### 3.2.1. DH as a Prognostic Factor


FEV_1_% decline—Data regarding the FEV_1_% at the time of the CPET and in the following 2 and 4 years after the test were available in all but one test, which was missing 4 years after the CPET. While the FEV_1_% did not change in the non-DH group (*p* = 0.213), a significant deterioration was observed in the DH group (*p* = 0.0034). In comparing the two groups, the FEV_1_% trend from the time of the CPET to 4 years after the test showed a substantial decline in the DH group compared to the non-DH group (*p* = 0.009, Figure 1). A correlation was found between the severity of DH and the FEV1%, with patients having lower ∆ICs also having lower FEV_1_% values (r = 0.36 and *p* = 0.019; Figure 2).



DH and LCI—LCI measurements were available for 26 tests. Significantly higher LCI values were found in the DH group, with a mean (SD) value of 17.45 (4.41) in the DH group versus 10.6 (4.40) in the non-DH group (*p* = 0.024, Figure 3). A negative correlation was found between the ∆IC and the LCI (r = −0.55 and *p* = 0.004; Figure 4).



DH and IV antibiotic courses—IV antibiotic courses due to pulmonary exacerbations during the four years following the CPET were documented in 13/25 (52%) in the non-DH group and in 12/16 (75%) in the DH group (*p* = 0.006). The total number of IV antibiotic courses was higher in the DH group—totaling 52 courses versus 35, respectively, with a *p* = 0.046. Furthermore, a correlation was found between the degree of DH and the frequency of pulmonary exacerbations (r = −0.43 and *p* = 0.005; Figure 5).


#### 3.2.2. Peak VO_2_ as a Prognostic Factor

A positive correlation was found between Peak VO_2_, measured at peak exercise, and the FEV_1_% 4 years after the CPET—meaning that a higher peak VO_2_ was found to predict a higher FEV_1_% after 4 years (r = 0.306 and *p* = 0.05; Figure 6). A negative correlation was found between the peak VO_2_ and LCI (r = −0.46 and *p* = 0.014), as well as between the peak VO_2_ and IV antibiotic courses due to pulmonary exacerbations (r = −0.46 and *p* = 0.014).

#### 3.2.3. End Tidal CO_2_ (EtCO_2_) and Tidal Volume (TV)/IC Ratio as Prognostic Factors

The EtCO_2_ values during exercise were similar between the dynamic hyperinflation group and the nondynamic hyperinflation group, with a mean (SD) of 33.92 (4.07) mmHg and 32.75 (5.82) mmHg, respectively (*p* = 0.52). The TV/IC ratios also did not differ, with mean (SD) ratios of 0.58 (0.30) in the dynamic hyperinflation group compared to 0.64 (0.17) in the nondynamic hyperinflation group (*p* = 0.40).

## 4. Discussion

In the present study, by summarizing the CPET results in a cohort of people with CF and examining the CPET parameters as prognostic factors, we showed that DH was found to correlate with lung function deterioration and pulmonary exacerbations measured by the need for IV antibiotic courses. DH was shown to be a common phenomenon in people with CF—with 39% of our cohort of patients demonstrating DH. During the CPETs, although DH was not associated with worse aerobic capacity, with no change in other respiratory measures at maximal effort, including peak VO_2_, patients with DH demonstrated a significantly higher breathing frequency that suggested less efficient ventilation. Nonetheless, peak VO_2_ was another predictor of pulmonary function in the 4-year follow-up.

Nixon et al. found that higher levels of aerobic fitness in patients with cystic fibrosis are associated with a significantly lower risk of dying and concluded that measurement of the peak VO_2_ appears to be valuable for predicting a prognosis [5]. Vendrusculo et al. have also recently found that low levels of peak VO_2_ are associated with an increase of 4.9-fold in the risk of mortality in people with CF [7]. This indicates that VO_2_ could be an important follow-up variable to measure in addition to the FEV_1_. On the other hand, other studies failed to show a correlation between the peak VO_2_ and the long-term prognosis [6]. Our results support the notion that peak VO_2_ is a prognostic factor in people with CF, with higher values predicting better lung function and reduced episodes of pulmonary exacerbations.

Lower IC at rest correlated with breathlessness and increased number of hospitalizations in the study by Vilozni et al. [17]. Furthermore, Stevens et al. showed that people with CF with static hyperinflation demonstrated a significant reduction in exercise performance [18].

When addressing DH, several studies [9,19] have shown that it is commonly seen in people with CF, but no association between DH and ventilatory limitation was demonstrated. Savi et al. [20] have also found DH to be prevalent in mild to moderate CF, and although their exercise tolerance was reduced, the daily physical activity was not impaired.

Conversely, other studies [10,11] did demonstrate a correlation between DH and reduced lung function, as well as exercise intolerance. Stevens et al. evaluated CPET parameters, including DH expression in 109 mild to moderate CF patients; they found a correlation between DH and reduced lung function parameters, including the FEV_1_%. They also found a correlation between DH and reduced CPET parameters such as oxygen uptake, minute ventilation, tidal volume, and work rate at peak exercise. In our study, we did not find differences in most respiratory exercise parameters in patients with DH, which is probably because our cohort included patients with better lung function than the patients in the study by Stevens et al. However, our study did demonstrate a significant deterioration in the FEV_1_% in DH patients compared to the non-DH group during the 4 years after the CPET. Stevens et al. did not find DH to have a prognostic value with regards to lung function deterioration in the following two years. These differences might be explained by the extended follow-up in our study, as well as the differences in the study population—with younger patients having been examined in our present study.

The LCI measurements in our study were positively correlated with DH. In a study by Chelabi et al. [21], LCI was evaluated as a possible marker of exercise limitation in children with mild CF, and although none of the children in the study had documented DH, and the end expiratory lung volumes were higher in patients with higher LCI values.

Pulmonary exacerbations treated with IV antibiotics during the four years following the CPETs were significantly more frequent in the DH group compared to the non-DH group. Furthermore, a positive correlation was found between the degree of dynamic hyperinflation and the frequency of pulmonary exacerbations. To our knowledge, this is the first study to examine the correlation between DH and pulmonary exacerbations as a marker of clinical deterioration in people with CF.

It is well known that in COPD the increase in the respiratory rate while exercising in the presence of expiratory flow limitation might cause DH [22]. Studies addressing bronchodilator therapy in COPD patients have demonstrated a rise in the tidal volume, which resulted in a decrease in the breathing frequency and a rise in exercise tolerance [22,23,24]. Reversible airway obstruction is common in CF, and it is more frequent in younger patients and in those with a severe genotype [25]. To date, no studies addressing bronchodilator therapy for dynamic hyperinflation in people with CF have been published; our results suggest that bronchodilator treatment might benefit people with CF demonstrating DH.

Our study has several limitations. First, the study is retrospective, although the CPET is part of routine follow up in our clinic. Second, our study group was small, as children under 10 years and patients with low FEV_1_ values did not perform this test. In addition, several tests were not technically acceptable as maximal tests. Moreover, two issues can confound the measurements of the inspiratory capacity: firstly, the plateau in the tidal volume could occur physiologically after the respiratory compensation point for lactic acidosis. Secondly, the inspiratory capacity maneuvers can potentially underestimate the magnitude of dynamic hyperinflation in patients with COPD who retain CO_2_ during exercise, because hypercapnia is associated with increasing neural respiratory drive. In order to eliminate the bias of CO_2_ retention, we added an analysis of the EtCO_2_ at maximal effort. The comparison of this parameter between the dynamic hyperinflation and nondynamic hyperinflation groups showed no difference. Lastly, we had more females in the DH group, which may be due to the fact that women present a more rapid deterioration in adolescence and young adulthood.

In conclusion, our study demonstrates that DH and peak VO_2_ are both associated with more frequent pulmonary exacerbations, as well as lung function deterioration, in people with CF. These results suggest that the VO_2_ in combination with DH might be used as an early marker to predict pulmonary deterioration in CF. Larger multi center prospective studies are needed to further evaluate the role of both markers in predicting lung function deterioration in people with CF.

## Figures and Tables

**Figure 1 jcm-12-05834-f001:**
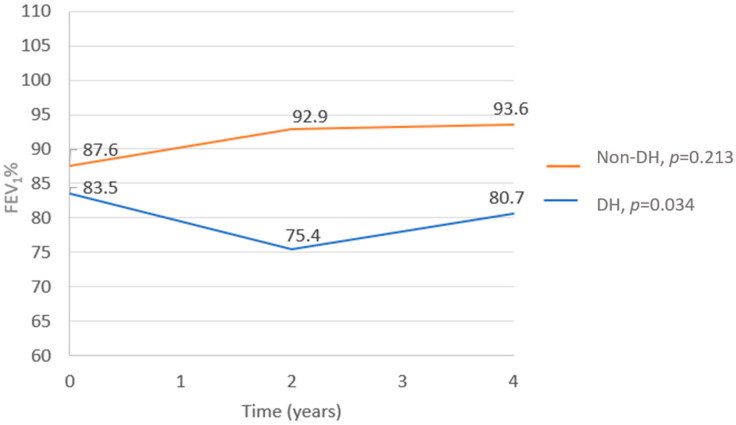
Median FEV_1_% trend in DH and non-DH groups from the time of CPETs to 4 years after the tests. General linear model; *p* = 0.009.

**Figure 2 jcm-12-05834-f002:**
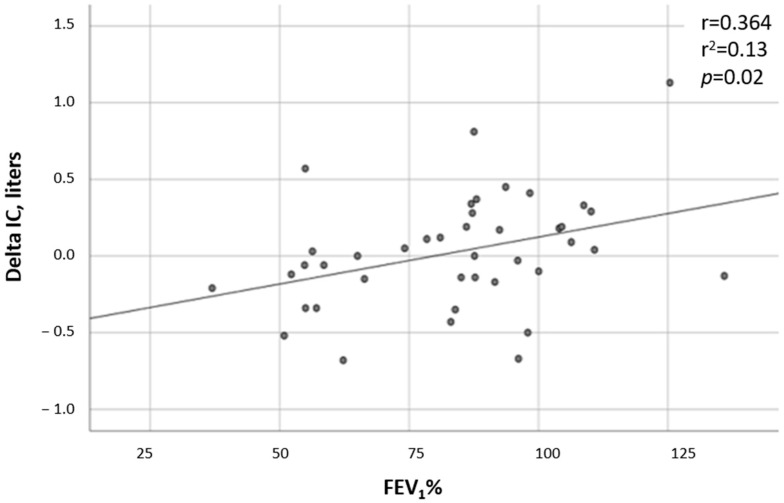
Correlation between ∆IC and predicted FEV_1_% predicted. Linear regression.

**Figure 3 jcm-12-05834-f003:**
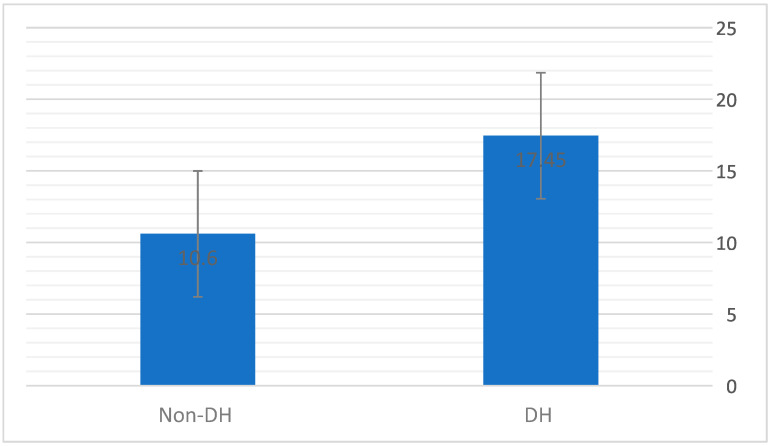
Mean LCI values in DH and non-DH groups. Mann–Whitney test.

**Figure 4 jcm-12-05834-f004:**
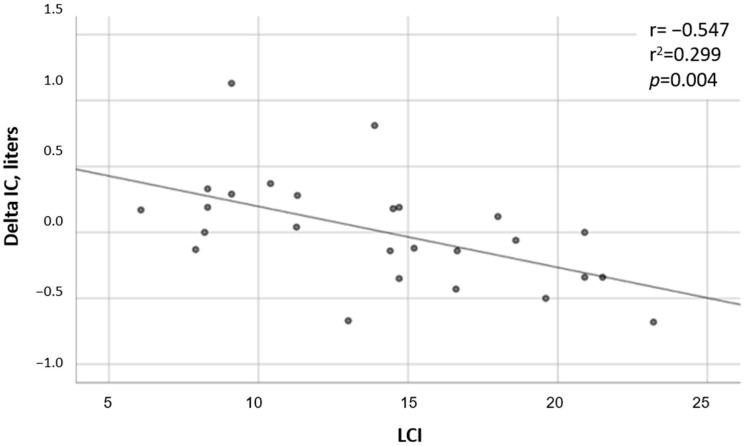
Correlation between ∆IC and LCI. Linear regression.

**Figure 5 jcm-12-05834-f005:**
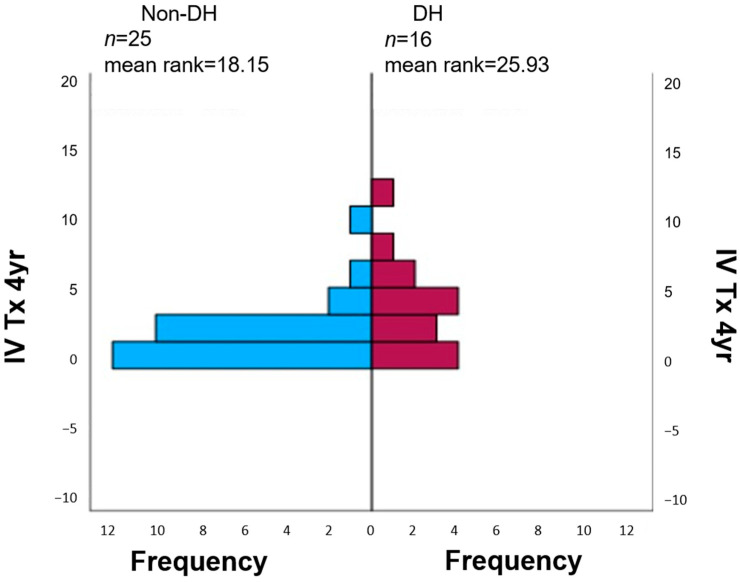
Independent sample Mann–Whitney U test for the number of IV courses in the four years after the CPETs in the DH and non-DH groups. Mann–Whitney test; IV antibiotic courses were documented in 13/25 (52%) patients in the non-DH group and in 12/16 (75%) in the DH group (*p* = 0.006), with a total number of IV antibiotic courses of 52 versus 35, respectively, and a *p* = 0.046.

**Figure 6 jcm-12-05834-f006:**
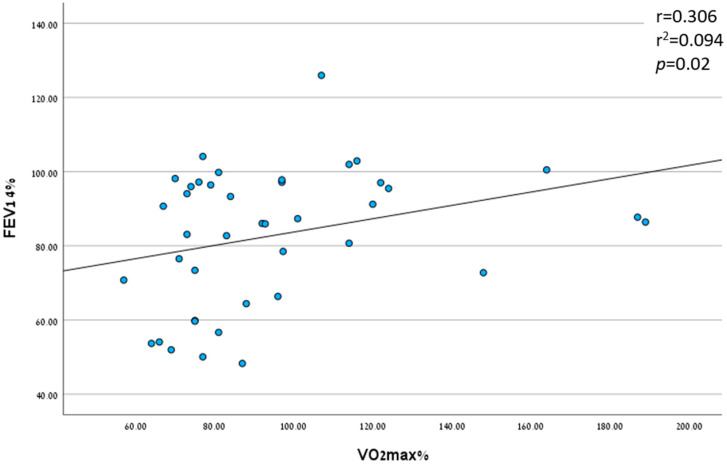
Correlation between FEV_1_% and Peak VO_2_. Linear regression.

**Table 1 jcm-12-05834-t001:** Patients characteristics.

	DH, *n* = 16	Non-DH, *n* = 25	*p*
Age—mean (SD) *	25.9 (14.8)	23.9 (10.2)	0.520
Female, *n* (%) ^#^	11 (69%)	8 (32%)	0.030
BMI *	21 (3.76)	21.8 (3.51)	0.490
Genotype ^#^ Both allele minimal function At least one allele residual function	10 (62.5%) 6 (37.5%)	15 (60%) 10 (40%)	0.873
Pancreatic insufficiency, *n* (%) ^#^	11 (69%)	17 (68%)	0.618
CF-related diabetes ^$^	3 (18%)	2 (8%)	0.362
CF-related liver disease ^$^	1 (6%)	6 (24%)	0.215
Chronic PSA ^#^	12 (75%)	14 (56%)	0.322
CFTR modulators tx, *n* (%) ^#^	5 (31%)	4 (16%)	0.276
FEV_1_%, median (SD) *	83.5 (25)	87.6 (19.2)	0.174

DH = dynamic hyperinflation; BMI = body mass index; PSA = Pseudomonas aeruginosa; * = independent samples *t*-test; ^#^ = Chi^2^ test; ^$^ = Fisher’s exact test.

**Table 2 jcm-12-05834-t002:** Cardiopulmonary exercise test (CPET) parameters ^#^.

	DH, *n* = 16	Non-DH, *n* = 25	*p*
Breathing frequency % of predicted breaths per minute, mean (SD) *	143.8 (34.6)	121.4 (30.6)	0.043
Breathing reserve %, mean (SD) *	25.8 (15.9)	28.6 (14.3)	0.580
Peak VO_2_%, mean (SD) *	97.3 (33.6)	92.8 (31.1)	0.668
VO_2_% predicted at the anaerobic threshold, mean (SD)	59.7 (22.4)	49.2 (19.1)	0.137

DH = dynamic hyperinflation; VO_2_ = oxygen consumption * = at peak exercise; ^#^ = independent samples *t*-test.

## Data Availability

The data presented in this study are available on request from the corresponding author. The data are not publicly available due to privacy restrictions.

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
