# Peer review of "Dynamic Hyperinflation While Exercising—A Potential Predictor of Pulmonary Deterioration in Cystic Fibrosis"

_jcm, 2023, doi:10.3390/jcm12185834_

Round 1

Reviewer 1 Report

The authors concluded that dynamic hyperinflation (DH) and peak V̇O2 were both associated with lung function deterioration and more frequent pulmonary exacerbations in pwCF.

* Can the authors give more explanation / argumentation why measuring of DH is important. What is the added value of DH?  

* Which (objective/subjective) citeria were used for a maximal CPET?

IC was measured by spirometry as patients were instructed to inspire fully after normal expiration and then to expire. DH was defined as a decrease of ≥5% in IC during maximal exercise.

* What is the evidence of this cut-off point?

 * Because measurements of exercise inspiratory capacity are not free from caveats, can the authors give a response on these caveats:

• the accrued knowledge comes from studies involving patients

with COPD ;

• it could be argued that in the absence of expiratory flow limitation,

vital capacity – not inspiratory capacity – is the actual limit for tidal

volume expansion;

• poor inspiratory capacity technique is common due to insufficient

familiarization with the maneuver;

• although a plateau in tidal volume frequently signals critically high inspiratory constraints it may occur physiologically after the respiratory compensation point for lactic acidosis;

• inspiratory capacity maneuvers can potentially underestimate the magnitude of dynamic hyperinflation in patients with COPD who retain CO2 during exercise because hypercapnia is associated with increasing neural respiratory drive;

• several measurements should be obtained throughout the test, avoiding misinterpretations based on a single inspiratory capacity at peak exercise; and

• given the potential underestimation of the flow reserves based on the maximal expiratory flow-volume loop, it might be difficult to judge the severity of expired flow limitation based on tidal-to-maximal flow-volume loop comparison. A change in the tidal expiratory limb’s morphology from convex to trapezoid might be helpful to suggest expiratory flow limitation.

It follows that a low/decreasing exercise inspiratory capacity – either due to dynamic gas trapping or inspiratory muscle weakness - predisposes to critically-high (i.e.,dyspnoea-generating) inspiratory constraints and “restrained breathing”, e.g., end-inspiratory lung volume/total lung capacity≥0.9, tidal volume/inspiratory capacity≥0.7, and a tidal volume plateau reached at an abnormally-low work rate.

* Can the authors give more information about these TV/IC ratio's and/or EILV/TLC ratio's?

N.A.

Author Response

Reviewer 1:

The authors concluded that dynamic hyperinflation (DH) and peak V̇O2 were both associated with lung function deterioration and more frequent pulmonary exacerbations in pwCF.

* Can the authors give more explanation / argumentation why measuring of DH is important. What is the added value of DH?  

We have now added to the introduction section a paragraph explaining the importance of dynamic hyperinflation measurement:

‘Inspiratory capacity (IC) tends to increase in healthy people while exercising, as recruited expiratory muscles ensure increased expiratory flow and decreased end expiratory lung volume. Dynamic hyperinflation is a phenomenon which appears in obstructive lung diseases such as chronic obstructive pulmonary disease (COPD) and CF while exercising. Collapse of airways during forced expiration in these obstructive lung diseases contributes to lung hyperinflation. Due to airway obstruction, exhalation may not be complete at the time the next breath is initiated, leading to increasing amounts of trapped air at end-exhalation and increasing end expiratory lung volume.  This process is termed dynamic hyperinflation.  Measuring inspiratory capacity is a surrogate measure of dynamic hyperinflation and increased end expiratory lung volume in obstructive lung disease. Assuming that total lung capacity remains constant, as exercise progresses and end expiratory lung volume increases, the inspiratory capacity decreases. Dynamic hyperinflation can limit ventilation during exercise. Work of breathing is increased, as the inspiratory muscles are at a mechanical disadvantage due to length-tension effects. At first, increased respiratory effort results in increased tidal volume. However, as exercise proceeds, there is a progressive decrease in IC without further increase in tidal volume. This results in the cardinal symptom of dynamic hyperinflation which is dyspnea on exertion. In healthy individuals, the exercise tidal flow volume loop (extFVL) and the maximal flow volume loop (MFVL) are distinct. However, as end expiratory lung volume increases with dynamic hyperinflation, there is encroachment of the extFVL on the MFVL’.

* Which (objective/subjective) citeria were used for a maximal CPET?

Thank you for this comment. We added a list of the applicability criteria used for a maximal CPET, by the ERS:

The criteria for acceptable CPET results included- a respiratory exchange ratio (RER) of ≥ 1.05 for adults and ≥ 1.03 for children; peak heart rate >100% predicted in adults or ≥195 bpm in children; ventilation at peak exercise exceeding Maximum voluntary ventilation (MVV); V̇O2 peak ≥100% predicted or maximal work rate ≥100% predicted.

IC was measured by spirometry as patients were instructed to inspire fully after normal expiration and then to expire. DH was defined as a decrease of ≥5% in IC during maximal exercise.

* What is the evidence of this cut-off point?

Since there are different criteria used in the literature, and no clear cut-off point for the definition of dynamic hyperinflation, we used the study by O’Donell et al, now sited in the article. In this study measuring dynamic hyperinflation in COPD patients there was a great variability, with a mean rise in inspiratory capacity of 4% in healthy people and a mean descent of 14% in COPD patients. 

 * Because measurements of exercise inspiratory capacity are not free from caveats, can the authors give a response on these caveats: the accrued knowledge comes from studies involving patients with COPD;

  • it could be argued that in the absence of expiratory flow limitation, vital capacity – not inspiratory capacity – is the actual limit for tidal volume expansion;

Thank you for this important point. We have added this explanation to the methods section:’ In the absence of expiratory flow limitation, vital capacity and not inspiratory capacity might be the actual limit for tidal volume expansion. It should therefore be noted that expiratory flow limitation was a major feature in our patient group. For this population, as inspiratory capacity is a surrogate measure for end expiratory lung volume, we chose to use this measure in our study’. 

  • poor inspiratory capacity technique is common due to insufficient familiarization with the maneuver;

We have now added to the methods section the following paragraph: ‘In order to practice the maneuver, the patients performed several IC maneuvers before commencing the CPET. In addition, during the CPET, maneuvers were repeated every 2-3 minutes and only technically acceptable tests at maximal exercise, as reviewed by an experienced technician- were included’.

  • although a plateau in tidal volume frequently signals critically high inspiratory constraints it may occur physiologically after the respiratory compensation point for lactic acidosis;
  • inspiratory capacity maneuvers can potentially underestimate the magnitude of dynamic hyperinflation in patients with COPD who retain CO2 during exercise because hypercapnia is associated with increasing neural respiratory drive;

Thank you for the comment, these are indeed two limitations of our study, and we have now added them into the discussion: ‘Two issues can confound measurements of inspiratory capacity: firstly, the plateau in tidal volume could occur physiologically after the respiratory compensation point for lactic acidosis. Secondly, inspiratory capacity maneuvers can potentially underestimate the magnitude of dynamic hyperinflation in patients with COPD who retain CO2 during exercise because hypercapnia is associated with increasing neural respiratory drive’.

In order to eliminate the bias of CO2 retention we added an analysis of EtCO2 at maximal effort. Comparison of this parameter between the dynamic hyperinflation and non-dynamic hyperinflation groups showed no difference:

  • End tidal CO2 (EtCO2) and tidal volume (TV)/IC ratio as prognostic factors

    EtCO2 values during exercise were similar between the dynamic hyperinflation group and the non-dynamic hyperinflation group, with mean (SD) of 33.92 (4.07) mmHg and 32.75 (5.82) mmHg, respectively (p=0.52).

  • several measurements should be obtained throughout the test, avoiding misinterpretations based on a single inspiratory capacity at peak exercise;

Please see above- regarding repeated IC maneuvers- we have added a paragraph to the methods section regarding this issue.

  • and given the potential underestimation of the flow reserves based on the maximal expiratory flow-volume loop, it might be difficult to judge the severity of expired flow limitation based on tidal-to-maximal flow-volume loop comparison. A change in the tidal expiratory limb’s morphology from convex to trapezoid might be helpful to suggest expiratory flow limitation.

Thank you for raising this very important point - as the morphology is a subjective observation, it is difficult to quantify it for study of our entire group.

It follows that a low/decreasing exercise inspiratory capacity – either due to dynamic gas trapping or inspiratory muscle weakness - predisposes to critically-high (i.e.dyspnoea-generating) inspiratory constraints and “restrained breathing”, e.g., end-inspiratory lung volume/total lung capacity≥0.9, tidal volume/inspiratory capacity≥0.7, and a tidal volume plateau reached at an abnormally-low work rate.

* Can the authors give more information about these TV/IC ratio's and/or EILV/TLC ratio's?

Thank you for this important comment. We have added new data regarding TV/IC ratios. Comparison of this parameter between the dynamic hyperinflation and non-dynamic hyperinflation groups showed no difference:

3.2.3.    End tidal CO2 (EtCO2) and tidal volume (TV)/IC ratio as prognostic factors

            EtCO2 values during exercise were similar between the dynamic hyperinflation group and the non-dynamic hyperinflation group, with mean (SD) of 33.92 (4.07) mmHg and 32.75 (5.82) mmHg, respectively (p=0.52). TV/IC ratios also did not differ – with mean (SD) ratios of 0.58 (0.30) in the dynamic hyperinflation group compared to 0.64 (0.17) in the non- dynamic hyperinflation group (p=0.40).

Reviewer 2 Report

Lung disease is the leading cause of death in cystic fibrosis (CF). In this paper, the authors investigated potential early diagnostic methods for pulmonary function decline in CF. Their findings suggest that dynamic hyperinflation (DH) could be a potential predictor for lung disease progression in CF. The findings of this paper are valuable to the field, but the authors should consider addressing the following comments to make their data more convincing.

Specific comments:

1.      The use of abbreviations that are not common in the field makes the article more difficult to read. The authors should use fewer abbreviations.

2.      The results suggested that dynamic hyperinflation could be a potential predictor of CF lung disease progression. The title of the paper and the first sentence of the Discussion session are too strong to be supported by the data.

3.      Since CF lung disease symptoms could be improved by medications (I am not expecting these patients to be on Ivacaftor or Trikafta). It is good to mention the routine medications that the CF patients were taking in the Method.

4.      The genotype of the DH patients in Table 1, I only see 10 (62.5%). Not sure whether it belongs to the “Both allele minimal function” or “At least one allele residual function”.

5.      In the tables and graphs, the authors should include the type of tests that were used for p value calculation.

6.      In several places, the authors described the findings including data or p value in the main text, but I cannot find the data in the graphs.

Examples:

1) “DH group compared to the non-DH group (p= 0.009, graph 1)”;

2) “ΔIC also having lower FEV1% values (r=0.36, p=0.019, graph 2)”;

3) “Significantly higher LCI values were found in the DH group, with a mean (SD) value of 17.45 (4.41) in the DH group versus 10.6 (4.40) in the non-DH group (p=0.024).”

4) “IV antibiotic courses due to pulmonary ex-acerbations during the four years following the CPET were documented in 13/25 (52%) in the non-DH group and in 12/16 (75%) in the DH group (p=0.006). The total number of IV antibiotic courses was higher in DH group- 52 courses versus 35, respectively, p=0.046. Furthermore, a correlation was found between the degree of DH and the frequency of pulmonary exacerba-tions (r=-0.43, p=0.005, graph 4)”.

5) Data related to the “3.2.2 Peak VO2 as a prognostic factor”.

7.      Graph 2 is not the correct graph.

Author Response

Reviewer 2:

Lung disease is the leading cause of death in cystic fibrosis (CF). In this paper, the authors investigated potential early diagnostic methods for pulmonary function decline in CF. Their findings suggest that dynamic hyperinflation (DH) could be a potential predictor for lung disease progression in CF. The findings of this paper are valuable to the field, but the authors should consider addressing the following comments to make their data more convincing.

Specific comments:

  1. The use of abbreviations that are not common in the field makes the article more difficult to read. The authors should use fewer abbreviations.

Thank you for this comment. We have changed the text accordingly.

  1. The results suggested that dynamic hyperinflation could be a potential predictor of CF lung disease progression. The title of the paper and the first sentence of the Discussion session are too strong to be supported by the

Thank you for this important comment. We have changed the title and the discussion accordingly- with the new title: ‘Dynamic Hyperinflation while Exercising- A Potential Predictor of Pulmonary Deterioration in Cystic Fibrosis’ and in the beginning of the of the discussion ‘…we showed that dynamic hyperinflation was found to correlate with lung’

  1. Since CF lung disease symptoms could be improved by medications (I am not expecting these patients to be on Ivacaftor or Trikafta). It is good to mention the routine medications that the CF patients were taking in the Method.

We have added a description of the routine mediations used by our CF patients to the methods section: ‘Routine medications used by the patients in our CF center include hypertonic saline and dornase alfa inhalations as well as inhaled and oral antibiotics, in accordance with bacteria found in sputum cultures. Routine treatment also includes physiotherapy, pancreatic enzyme replacement therapy and vitamins. As previously mentioned, some patients received CFTR modulator treatment during the study perio’- with no difference between the 2 study groups

  1. The genotype of the DH patients in Table 1, I only see 10 (62.5%). Not sure whether it belongs to the “Both allele minimal function” or “At least one allele residual function”.

We are sorry for the misunderstanding, and in the revised manuscript it is now clear that 10 (62.5%) belongs to the ‘Both allele minimal function’ group.

  1. In the tables and graphs, the authors should include the type of tests that were used for p value calculation.

We have now included a description of the types of tests used for calculation of p value in the tables and graphs.

  1. In several places, the authors described the findings including data or p value in the main text, but I cannot find the data in the graphs.

Examples:

1) “DH group compared to the non-DH group (p= 0.009, graph 1)”; 

We have now added the p value to legend of graph 1.

2) “ΔIC also having lower FEV1% values (r=0.36, p=0.019, graph 2)”;

Graph 3 appears twice instead of the correct graph 2, and was now replaced.    

3) “Significantly higher LCI values were found in the DH group, with a mean (SD) value of 17.45 (4.41) in the DH group versus 10.6 (4.40) in the non-DH group (p=0.024).” these results are not presented in any graph/table.

We have now added a new graph 3 to the article.

4) “IV antibiotic courses due to pulmonary ex-acerbations during the four years following the CPET were documented in 13/25 (52%) in the non-DH group and in 12/16 (75%) in the DH group (p=0.006). The total number of IV antibiotic courses was higher in DH group- 52 courses versus 35, respectively, p=0.046. Furthermore, a correlation was found between the degree of DH and the frequency of pulmonary exacerbations (r=-0.43, p=0.005, graph 4)”.

These results were only partially presented in graph 4 and we have now included them in the graph and in the legend.

5) Data related to the “3.2.2 Peak VO2 as a prognostic factor”.

We have now added graph 6, presenting the correlation between FEV1% and VO2 peak. Since our main aim was to evaluate the correlation between dynamic hyperinflation and pulmonary deterioration and since there are now 6 graphs and 2 tables in the article, we did not add more graphs, depicting the correlation between peal VO2 and other parameters. If the reviewers believe that more graphs should be added we will gladly do so.

  1. Graph 2 is not the correct graph.

We apologize for this mistake. Graph 2 was replaced

Round 2

Reviewer 2 Report

The authors have addressed most of my comments. 

Minor suggestions:

1. My previous comment regarding fewer abbreviations. I meant some abbreviations might break the flow of reading because the readers cannot remember all the uncommon abbreviations. In fact, I am OK about the "Dynamic hyperinflation" terms as "DH". The readers can remember it because it shows up again and again. But for the abbreviations like pwCF, CFRD, CFLD. They show up several times in the text, which may not be necessary to use the abbreviations. In addition, a few abbreviations in the Abstract are not defined for the first time. The author might consider either define them or not use those abbreviations.

2. Graph 5. The author might consider change the fond of the graph.

Author Response

Thank you for these important comments.

  1. We have changed all abbreviations accordingly in the main text and defined the relevant abbreviations in the abstract.
  2. We have replaced the font in graph 5.